# QE-BEV: Query Evolution for Bird's Eye View Object Detection in Varied Contexts

Jiawei Yao*
School of Engineering and
Technology, University of Washington
Tacoma, United States
jwyao@uw.edu

Yingxin Lai*
Department of Artificial Intelligence,
Xiamen University
Xiamen, China
laiyingxin2@gmail.com

Hongrui Kou*
Department of Vehicle Engineering,
Jilin University
Changchun, China
kouhr23@mails.jlu.edu.cn

Tong Wu
School of Engineering and
Technology, University of Washington
Tacoma, United States
tongwu0822@gmail.com

Ruixi Liu[†]
Department of Computer Science and
Engineering, Yonsei University
Seoul, Republic of Korea
handsomeroy@yonsei.ac.kr

## Abstract

3D object detection plays a pivotal role in autonomous driving and robotics, demanding precise interpretation of Bird's Eye View (BEV) images. The dynamic nature of real-world environments necessitates the use of dynamic query mechanisms in 3D object detection to adaptively capture and process the complex spatio-temporal relationships present in these scenes. However, prior implementations of dynamic queries have often faced difficulties in effectively leveraging these relationships, particularly when it comes to integrating temporal information in a computationally efficient manner. Addressing this limitation, we introduce a framework utilizing dynamic query evolution strategy, harnesses K-means clustering and Top-K attention mechanisms for refined spatio-temporal data processing. By dynamically segmenting the BEV space and prioritizing key features through Top-K attention, our model achieves a real-time, focused analysis of pertinent scene elements. Our extensive evaluation on the nuScenes and Waymo dataset showcases a marked improvement in detection accuracy, setting a new benchmark in the domain of query-based BEV object detection. Our dynamic query evolution strategy has the potential to push the boundaries of current BEV methods with enhanced adaptability and computational efficiency. Project page: https://github.com/Jiawei-Yao0812/QE-BEV

## CCS Concepts

• **Computing methodologies → Computer vision**; **3D imaging**; **Object detection**.

## Keywords

3D Object Detection, Bird's Eye View images, Dynamic Query Evolution, Computational Efficiency

---

*These authors contribute equally to this work.
[†]Corresponding author.

---

*MM '24, October 28-November 1, 2024, Melbourne, VIC, Australia*
© 2024 Copyright held by the owner/author(s).
ACM ISBN 979-8-4007-0686-8/24/10
https://doi.org/10.1145/3664647.3680807

**ACM Reference Format:**
Jiawei Yao, Yingxin Lai, Hongrui Kou, Tong Wu, and Ruixi Liu. 2024. QE-BEV: Query Evolution for Bird's Eye View Object Detection in Varied Contexts. In *Proceedings of the 32nd ACM International Conference on Multimedia (MM '24), October 28-November 1, 2024, Melbourne, VIC, Australia.* ACM, New York, NY, USA, 9 pages. https://doi.org/10.1145/3664647.3680807

## 1 Introduction

3D object detection is a pivotal task in various applications [3, 4] like autonomous driving, robotics, and surveillance [9, 13, 22]. In the field of 3D object detection, BEV (Bird's Eye View) algorithms [7, 15, 36] have gained increasing prominence due to their ability to provide a top-down perspective, simplifying complex 3D scenes [35] into 2D representations. This perspective aids in reducing computational complexity and enhancing the clarity of object localization. However, traditional query-based BEV methods have mainly relied on static queries [20, 21, 32], where the query weights are learned during the training phase and remain fixed during inference. This static nature restricts the model's capability to effectively leverage both spatial and temporal contexts and adapt to complex scenes. While recent studies like CMT [34] and UVTR [12] have introduced dynamic queries. The dynamic queries enable the model with the ability to adaptively capture complex spatial-temporal relationships. However, these methods typically employ a position-guided query mechanism, interacting simply with features or tokens through basic methods like adding position encoding or direct concatenation. This limits their ability to explore unknown areas in space, particularly in complex or dynamically changing environments. Figure 1 illustrates the limitations of static query-based methods, such as DETR3D [32], where queries are learnable during training but remain fixed during inference. This inflexibility hampers the model's responsiveness to dynamic changes within the scene. Existing dynamic query methods, while addressing some of these issues, often rely on simple position-guided interactions that do not fully capture the complexities of spatial and temporal variations in real-world environments. In contrast, our method introduces a dynamic querying mechanism that allows for queries to adapt to the input data iteratively.

In this vein, we introduce QE-BEV, a novel method that pioneers the use of advanced dynamic queries in query-based 3D object detection, pushing the boundaries of what can be achieved with BEV

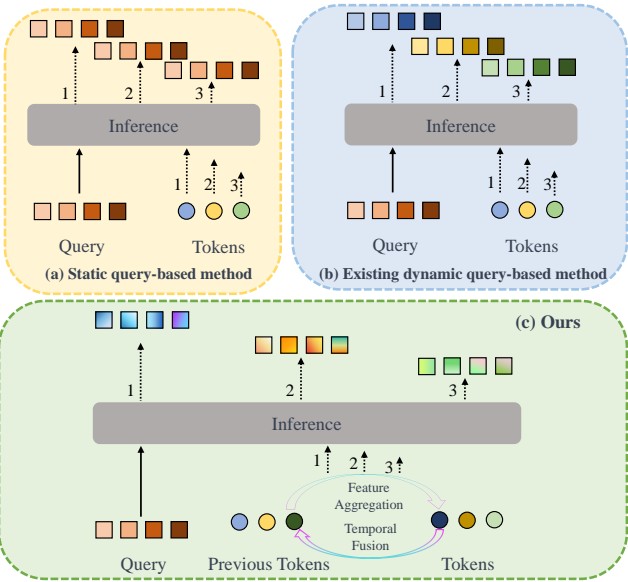

**Figure 1: Comparison of Different Query Methods. (a) Static query-based method: Queries are pre-defined and unchanging during inference, linking to a consistent set of tokens. (b) Existing dynamic query-based method: Queries are adaptive, updating their association with tokens, yet within a limited, position-guided context. (c) Our approach: Fusion of dynamic queries with both feature aggregation and temporal fusion, enabling queries to adapt more comprehensively by considering previous tokens, thereby capturing intricate object dynamics and relationships over time.**

(Bird's Eye View) algorithms. QE-BEV fundamentally redefines query dynamics through iterative adaptations, leveraging feature clustering for adaptive scene representation and employing a Top-K Attention mechanism that intelligently adjusts to the most relevant top-k feature clusters. This approach enables each query to flexibly and efficiently aggregate information across both proximal and distal feature clusters, enhancing the system's ability to interpret complex, multi-dimensional scenes.

In summary, our contributions are as follows:

- We introduce an innovative dynamic querying mechanism in QE-BEV, which utilizes Top-K Attention to dynamically focus on the most relevant feature clusters, enhancing the model's accuracy and adaptability to complex scenarios.
- We incorporate a Diversity Loss within the Top-K Attention framework to ensure that attention is not only paid to the most dominant features but also to less prominent ones, boosting the robustness of our model across varied environments.
- Our QE-BEV includes a Lightweight Temporal Fusion Module (LTFM) that significantly reduces computational overhead by reusing dynamic queries and their associated feature clusters, thereby streamlining the incorporation of temporal context. This model has been rigorously evaluated on the nuScenes [1] and Waymo [25] datasets, demonstrating significant improvements over state-of-the-art methods in terms of both accuracy and efficiency.

## 2 Related Work

**Query-based Object Detection in 2D and 3D.** Query-based object detection has gained significant advancements thanks to the introduction of the Transformer architecture [27]. Primary works like DETR [2] adopted a static query-based approach where queries are used to represent potential objects but do not adapt during the detection process. Various works [5, 26, 38] have focused on accelerating the convergence or improving the efficiency of these static query-based methods. However, these models, even when extended to 3D space [20, 32], inherently lack the ability to adapt queries to complex spatial and temporal relationships within the data. Our work diverges from this static paradigm by introducing dynamic queries that iteratively adapt during detection, effectively constituting a new paradigm in query-based object detection.

**Monocular and Multiview 3D Object Detection.** Monocular 3D object detection [24, 30, 31] and multiview approaches [9, 23] have been widely studied for generating 3D bounding boxes from 2D images. While effective, these methods generally operate under a static framework where features are extracted and used without further adaptation. Our work, QE-BEV, enhances this by dynamically adapting the queries in BEV space to capture both local and distant relationships, thus presenting a novel approach in the realm of 3D object detection.

**Static vs. Dynamic Paradigms in BEV Object Detection.** BEV-based object detection has progressed significantly, with methods typically employing a static approach where queries or feature representations remain constant during the detection process, as seen in DETR3D [32] and PETR series [20, 21]. These approaches, however, do not fully account for the spatial-temporal dynamics of real-world scenes. Recent works have introduced dynamic queries to address this limitation, such as MV2D [33], SimMOD [37], CMT [34], and UVTR [12], which allow for updates in response to input data. Our work extends these concepts by incorporating a dynamic paradigm where queries adapt iteratively, enhancing the model's ability to capture complex and changing spatial-temporal relationships in 3D object detection.

**Temporal Information in Object Detection.** Incorporating temporal information has been explored in various works [15, 19, 22]. However, these methods often introduce significant computational complexity and are constrained by the static nature of their query or feature representations. Our Lightweight Temporal Fusion Module (LTFM) not only efficiently integrates temporal context but does so in a dynamic manner, further emphasizing the shift towards a dynamic paradigm in 3D object detection.

## 3 Method

In this section, we introduce QE-BEV, a novel method designed for effective and efficient 3D object detection. Traditional static query-based methods lack the dynamism required to capture the diverse nature of 3D spaces. In contrast, QE-BEV harnesses dynamic queries that undergo iterative updates, and thereby achieves unparalleled adaptability in discerning diverse object attributes. The key components of QE-BEV are shown in Figure 2.

QE-BEV is composed of multiple integral components that synergize to facilitate robust and precise 3D object detection. The framework includes a backbone network responsible for initial feature

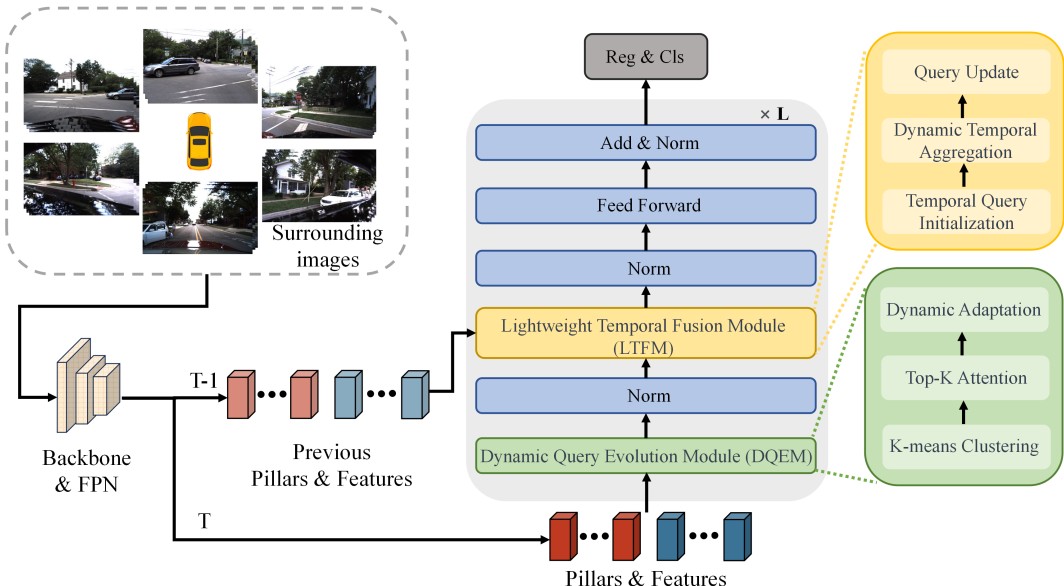

**Figure 2: The architecture of QE-BEV. Beginning with feature extraction from surrounding images using a backbone network and FPN, the architecture leverages previous pillars and features for temporal context. These are processed through the Dynamic Query Evolution Module (DQEM) for adaptive query refinement using K-means clustering and Top-K Attention. The Lightweight Temporal Fusion Module (LTFM) then integrates temporal information before the final query update, which combines dynamic temporal aggregation with the initial temporal query initialization. Finally, the updated queries are used for 3D object prediction.**

extraction. With the extracted feature, a Dynamic Query Evolution Module (DQEM) comes into play. First, DQEM exploits K-means clustering to groups features around each query, which brings adaptive structure representation for complex 3D scenarios. Afterwards, a Top-K Attention module is employed by DQEM to iteratively refine queries with their associated feature clusters. Finally, a Lightweight Temporal Fusion Module (LTFM) is incorporated to efficiently capture temporal context for each query.

## 3.1 Dynamic Query Evolution Module (DQEM)

**Initialization of Queries (Pillars).** In the context of 3D object detection, the initialization of queries plays a pivotal role in the subsequent detection performance. In the BEV space, these queries, often referred to as "pillars", serve as reference points or anchors that guide the detection process. The query set $Q$ can be represented as:

$$Q = \{(x_i, y_i, z_i, w_i, l_i, h_i, \theta_i, v_{x_i}, v_{y_i})\} \tag{1}$$

where $(x_i, y_i, z_i)$ is the spatial coordinates of the $i$-th pillar, indicating its position in the BEV space. $w_i, l_i, h_i$ are width, length and height of the pillar, respectively, providing the shape attributes. $\theta_i$ is the orientation angle of the pillar, offering insights into its alignment in the BEV space. $v_{x_i}$ and $v_{y_i}$ are velocity components of the pillar, capturing its motion dynamics.

In traditional methods like SparseBEV [19], these queries and their associate features are initialized based on pre-defined grid structures and remain static throughout the detection process. Such static nature are designed to capture general object patterns but is not adept at handling diverse scenarios with complex intricate object details.

On the contrary, in QE-BEV, the associated feature are grouped into a clustered structure, which well adapts to the complex 3D scene, and each pillar iteratively adjusts its attributes (like position, dimensions, or orientation) based on the associated feature clusters. Such dynamism renders the pillars better adaptability to the object attributes in the 3D scenes, leading to a more accurate and robust detection.

**K-means Clustering.** In QE-BEV, K-means clustering is first employed to divide the surrounding features $F$ of each query into $K$ clusters $C_1, \ldots, C_K$. The surrounding features refer to the features within a predefined spatial neighborhood of each query point, determined by a fixed radius $r$. These features include information such as geometry, texture, and color, and are extracted from multi-view data at the same time step.

The rationale behind employing K-means clustering lies in its ability to partition the feature space into clusters within which the feature variance is minimized. This enable each query to focus on groups of coherent features rather than unorganized points, which is a more adaptive and structured representation, thereby enhancing the model's ability to discern the objects in 3D scenes. After K-means clustering, each query $q$ will have an associated set of feature clusters $C_k$, formally:

$$C_k = \{f_i \mid c_i = k\}, \tag{2}$$

and the cluster center:

$$\mu_k = \frac{1}{|C_k|} \sum_{f_i \in C_k} f_i. \tag{3}$$

These clusters encapsulate the local patterns around each query, and provide the model with a more adaptive structured representation of the dynamic 3D scenes, serving as the foundation for the subsequent Top-K Attention steps.

**Top-K Attention Aggregation.** To allow each query to aggregate features in a dynamic way, we introduce a Top-K Attention mechanism. For each query $q$, we compute the attention weights over its associated feature clusters $C_k$ obtained from K-means clustering.

**Compute Attention Scores.** For each query feature $q$ and each cluster $C_k$, compute an attention score.

$$A_k = (W_q q)^T \cdot W_k \mu_k \qquad (4)$$

Here, $W_q$ represents the weight vector for the query and $W_k$ represents the weight vector for the cluster. The dot product measures the relevance between the query and each cluster.

This step allows the model to measure the importance of each feature cluster with respect to the query, enabling more informed aggregations.

**Select Top-K Clusters.** Sort the attention scores $A_k$ in descending order and select the top-K clusters.

$$\text{Top-K clusters} = \text{argmax}_k(A_k), \quad k = 1, \dots, K \qquad (5)$$

This selective attention mechanism enables each query to focus on the most relevant clusters, which may even be farther away, thus enriching the aggregated feature.

**Weighted Feature Aggregation.** Aggregate the selected clusters using their attention weights to form the aggregated feature $q'$ to update each query $q$.

$$q' = \sum_{k \in \text{Top-K}} \text{Softmax}(A)_k \cdot \mu_k \qquad (6)$$

The weighted sum allows for a rich combination of features, enabling each query to adaptively focus on different aspects of the surrounding features.

The aggregated feature $q'$ serves as the foundation for 3D object prediction. By allowing each query to aggregate information even from distant clusters, the model's capacity to capture long-range dependencies is significantly enhanced. Such capacity is particularly crucial in 3D object detection, where objects might have parts that are spatially separated but are contextually related.

**Diversity Loss for Balanced Feature Aggregation.** The proposed Top-K Attention mechanisms has the risk of focusing excessively on the most relevant features corresponding to each query. While this approach is effective in capturing dominant patterns, it often neglects the long-tail or less prominent features that could be critical for certain edge cases or specific scenarios. For example, in a 3D object detection task involving vehicles and pedestrians, focusing solely on the most relevant features might capture the overall shape of a vehicle but miss out on smaller but important details like side mirrors or indicators, which are essential for precise localization and classification.

To address this limitation, we introduce a Diversity Loss $L_{\text{div}}$. This loss function aims to balance the attention mechanism by ensuring that not only the most relevant but also the less prominent

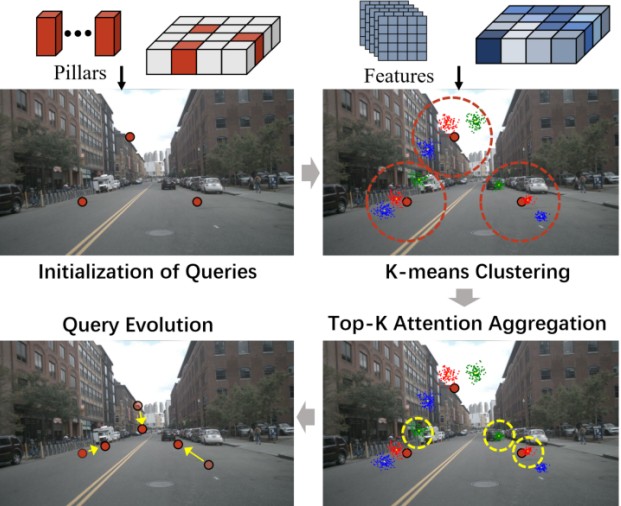

**Figure 3: Dynamic Query Evolution Module (DQEM). The sequence begins with the initialization of query pillars, which are then spatially coordinated in the BEV space based on extracted features. Subsequent K-means clustering organizes these features into distinct clusters. The process continues with Top-K Attention Aggregation, dynamically refining each query based on the most informative feature clusters. This results in an evolved query set adept at capturing the complex, multi-dimensional relationships.**

features are considered. Unlike conventional entropy-based losses, which are agnostic to the task at hand, our Diversity Loss is meticulously crafted for 3D object detection, ensuring a balanced attention distribution across different feature clusters, formally:

$$L_{\text{div}} = -\sum_{k=1}^{K} p_k \log p_k, \qquad (7)$$

where the following function serves as a critical component for stabilizing the gradient flow during the back-propagation process, especially when dealing with clusters of varying relevance:

$$p(k) = \frac{\exp(A_k)}{\sum_{j=1}^{K} \exp(A_j)}. \qquad (8)$$

This Diversity Loss brings several advantages. Firstly, it promotes a balanced feature representation by encouraging the model to pay attention to a variety of features, not just the most prominent ones. This is particularly useful for capturing less obvious but potentially crucial features. Secondly, the approach enhances the model's robustness, allowing it to adapt better to different scenarios and noise levels. Lastly, it fosters a more comprehensive understanding of the data, thereby improving the model's generalization capabilities.

**Dynamic Adaptation of Queries.** After initializing the queries as pillars and performing K-means clustering to obtain feature clusters $C_k$, the next crucial step is dynamically adapting these queries based on the Top-K Attention mechanism. This dynamic adaptation is the key difference from SparseBEV, where the queries are static. In QE-BEV, each query not only captures local information but also

dynamically updates itself to aggregate relevant features from a large scope of feature clusters.

**Initial Feature Aggregation.** For each query $q$, aggregate the initial set of features using a simple average or any other aggregation method.

$$q \leftarrow \frac{1}{|F|} \sum_{f \in F} f \qquad (9)$$

This initial aggregation serves as a baseline, capturing the immediate vicinity of the query. It acts as an anchor, grounding the subsequent dynamic adaptations.

**Top-K Attention Update.** Apply the previously described Top-K Attention mechanism to adaptively update each query $q$ using its associated feature clusters $C_k$.

$$q \leftarrow q' + \beta \cdot q \qquad (10)$$

Here, $q'$ is the aggregated feature obtained from Top-K Attention, and $\beta$ is a hyper-parameter that controls the blending of initial and dynamically aggregated features.

This step allows each query to adaptively refine its feature representation based on both local and long-range information, enhancing its ability to capture complex patterns and relationships.

**Iterative Update.** Repeat the K-means clustering and Top-K Attention steps, using the newly updated queries $q$ as the new pillars for the next iteration. Such iterative update ensures the queries continuously adapting to the varying feature landscape, thereby increasing the model's robustness and adaptability.

By iteratively updating queries through a combination of K-means clustering and Top-K Attention, QE-BEV ensures each query is both locally and globally informed, thereby capturing richer and more balanced feature representations. This dynamic adaptation is a significant advancement over SparseBEV, where pillars remain static and cannot adapt to capture long-range dependencies.

## 3.2 Lightweight Temporal Fusion Module

In QE-BEV, the key advantage of our Lightweight Temporal Fusion Module (LTFM) lies in its computational efficiency. Unlike traditional temporal fusion methods that rely on resource-intensive recurrent or convolutional layers, LTFM leverages the already computed dynamic queries $Q$ and their corresponding feature clusters $C_k$, thereby avoiding additional heavy computations.

**Temporal Query Initialization.** The temporal queries $q$ are initialized using a weighted combination of current and previous dynamic queries , thus reusing existing computations.

$$q \leftarrow \alpha \cdot q + (1 - \alpha) \cdot q_{\text{previous}} \qquad (11)$$

By reusing the dynamic queries, we eliminate the need for separate temporal query extraction, thereby reducing computational overhead.

**Dynamic Temporal Aggregation.** The Top-K Attention mechanism is applied directly to $q$, reusing the previously computed feature clusters $C_k$ for both current and previous time steps.

$$q' = \text{Top-K Attention}(q, F_{\text{current}}, F_{\text{previous}}) \qquad (12)$$

This obviates the need for separate temporal feature extraction, further reducing computational cost.

**Table 1: The FPS of QE-BEV and the baseline methods are measured with NVIDIA RTX 3090 GPU. All the methods use ResNet 50 as the backbone, with the same input resolution 704 × 256 and query number 900.**

| Method | Backbone | Input Size | FPS |
|---|---|---|---|
| SparseBEV [19] | ResNet50 | 704×256 | 15.4 |
| StreamPETR [28] | ResNet50 | 704×256 | 25.9 |
| QE-BEV | ResNet50 | 704×256 | 35.6 |

**Query Update.** The temporal queries $q$ are updated using the aggregated temporal features $q'$, similar to the dynamic query update in the previous sections.

$$q \leftarrow q' + \beta \cdot q \qquad (13)$$

The update operation is computationally light, as it only involves basic arithmetic operations, thus bringing the computational efficiency.

LTFM provides an efficient way to incorporate temporal context without introducing a significant computational burden. By reusing existing computations to avoid additional complex operations, LTFM offers a lightweight yet effective solution for temporal fusion.

## 3.3 Computational Complexity

The computational efficiency of QE-BEV is one of its key advantages. Below, we quantify this in terms of time complexity: The overall time complexity is approximately $O(nKId + n \log n + n)$, where $n$ is the number of data points, $K$ is the number of cluster centers, $I$ is the number of iterations in K-means, $d$ is the dimensionality of each data point. This is relatively low compared to methods that require more complex temporal fusion techniques such as RNNs or CNNs. As shown in Table 1, QE-BEV achieves considerable higher efficiency than two common baselines.

## 4 Experiment
## 4.1 Implementation Details

We adopt ResNet [6] as the backbone, the temporal module in our model is designed to be lightweight and we use a total of $T = 8$ frames by default, with an interval of approximately 0.5s between adjacent frames. For label assignment between ground-truth objects and predictions, we use the Hungarian algorithm [10]. The loss functions employed are focal loss [16] for classification and L1 loss for 3D bounding box regression, augmented by our custom Diversity Loss $L_{\text{div}}$ with a weight factor of $\lambda = 0.1$. The initial learning rate is $2 \times 10^{-4}$, and it is decayed using a cosine annealing policy. In line with recent advancements, we adjust the loss weight of $x$ and $y$ in the regression loss to 2.0, leaving the others at 1.0, to better capture spatial intricacies. We also incorporate Query Denoising to stabilize training and speed up convergence, as suggested by the recent work StreamPETR [28]. For our K-means clustering, $K$ is set to 6. The number of Top-K clusters for attention is set to 4. The hyperparameter $\beta$ used for blending in query update is set to 0.6, and $\alpha$ for temporal fusion in the Lightweight Temporal Fusion Module (LTFM) is set to 0.4.

**Table 2: Performance comparison on nuScenes `val` split. † benefits from perspective pretraining.**

| Method | Backbone | Input Size | Epochs | NDS | mAP | mATE | mASE | mAOE | mAVE | mAAE |
|---|---|---|---|---|---|---|---|---|---|---|
| PETRv2 [21] | ResNet50 | 704 × 256 | 60 | 45.6 | 34.9 | 0.700 | 0.275 | 0.580 | 0.437 | 0.187 |
| CMT [34] | ResNet50 | 704 × 256 | 60 | 46.0 | 40.6 | – | – | – | – | – |
| UVTR [12] | ResNet50 | 704 × 256 | 60 | 47.2 | 36.2 | 0.756 | 0.276 | 0.399 | 0.467 | 0.189 |
| BEVStereo [11] | ResNet50 | 704 × 256 | 90 | 50.0 | 37.2 | 0.598 | 0.270 | 0.438 | 0.367 | 0.190 |
| BEVPoolv2 [8] | ResNet50 | 704 × 256 | 90 | 52.6 | 40.6 | 0.572 | 0.275 | 0.463 | 0.275 | 0.188 |
| SOLOFusion [22] | ResNet50 | 704 × 256 | 90 | 53.4 | 42.7 | 0.567 | 0.274 | 0.511 | 0.252 | 0.181 |
| Sparse4Dv2 [18] | ResNet50 | 704 × 256 | 100 | 53.9 | 43.9 | 0.598 | 0.270 | 0.475 | 0.282 | 0.179 |
| StreamPETR † [28] | ResNet50 | 704 × 256 | 60 | 55.0 | 45.0 | 0.613 | 0.267 | 0.413 | 0.265 | 0.196 |
| SparseBEV [19] | ResNet50 | 704 × 256 | 36 | 54.5 | 43.2 | 0.619 | 0.283 | 0.396 | 0.264 | 0.194 |
| SparseBEV †[19] | ResNet50 | 704 × 256 | 36 | 55.8 | 44.8 | 0.595 | 0.275 | 0.385 | 0.253 | 0.187 |
| **QE-BEV** | ResNet50 | 704 × 256 | 60 | 56.1 | 45.4 | 0.601 | 0.272 | 0.381 | 0.235 | 0.168 |
| **QE-BEV †** | ResNet50 | 704 × 256 | 60 | **57.8** | **46.9** | 0.577 | 0.264 | 0.353 | 0.235 | 0.187 |
| DETR3D † [32] | ResNet101 | 1600 × 900 | 24 | 43.4 | 34.9 | 0.716 | 0.268 | 0.379 | 0.842 | 0.200 |
| UVTR [12] | ResNet101 | 1600 × 900 | 24 | 48.3 | 37.9 | 0.731 | 0.267 | 0.350 | 0.510 | 0.200 |
| BEVFormer † [15] | ResNet101 | 1600 × 900 | 24 | 51.7 | 41.6 | 0.673 | 0.274 | 0.372 | 0.394 | 0.198 |
| BEVDepth [13] | ResNet101 | 1408 × 512 | 90 | 53.5 | 41.2 | 0.565 | 0.266 | 0.358 | 0.331 | 0.190 |
| Sparse4D † [17] | ResNet101 | 1600 × 900 | 48 | 55.0 | 44.4 | 0.603 | 0.276 | 0.360 | 0.309 | 0.178 |
| SOLOFusion [22] | ResNet101 | 1408 × 512 | 90 | 58.2 | 48.3 | 0.503 | 0.264 | 0.381 | 0.246 | 0.207 |
| SparseBEV † [19] | ResNet101 | 1408 × 512 | 24 | 59.2 | 50.1 | 0.562 | 0.265 | 0.321 | 0.243 | 0.195 |
| **QE-BEV †** | ResNet101 | 1408 × 512 | 24 | **61.1** | **51.7** | 0.568 | 0.248 | 0.344 | 0.229 | 0.193 |

## 4.2 Datasets and Evaluation Criteria

Our experiments utilize the nuScenes dataset [1], a rich source of multi-modal sensor information encompassing 1000 driving sequences, each lasting around 20s. Annotations are available at a rate of 2 Hz for key frames. Each frame in the dataset offers a comprehensive 360-degree field of view through six camera sensors. For the task of 3D object detection, the dataset incorporates approximately 1.4 million 3D bounding boxes across 10 categories of objects.

We adopt a similar task setting as in previous works [19] for Birds-Eye View (BEV) segmentation. The official evaluation metrics of nuScenes are comprehensive; they not only include mean Average Precision (mAP), which is calculated based on the center distance in the ground plane instead of 3D IoU, but also feature five additional True Positive (TP) error metrics: ATE, ASE, AOE, AVE, and AAE, to measure the errors in translation, scale, orientation, velocity, and attributes respectively. To provide a unified score that captures multiple facets of detection performance, the nuScenes Detection Score (NDS) is used, defined as:

$$NDS = \frac{1}{10}\left[5 \times mAP + \sum_{mTP \in TP}(1 - \min(1, mTP))\right] \quad (14)$$

In addition, we verify the advantage of QE-BEV on the Waymo dataset [25], with the same experiment settings exploited in [28].

## 4.3 Comparison with the State-of-the-art Methods

Table 2 presents the performance of our QE-BEV on the nuScenes validation dataset, compared with other state-of-the-art methods, which outperforms all other methods by a considerable margin. With a ResNet50 backbone and an input size of 704 × 256, QE-BEV achieves a nuScenes Detection Score (NDS) of 56.1, which is higher than the 54.5 achieved by SparseBEV. More significantly, when perspective pre-training is applied, indicated by the † symbol,

**Table 3: Performance Comparison on Waymo `val` set.**

| Methods | Backbone | mAPL↑ | mAP↑ | mAPH↑ |
|---|---|---|---|---|
| BEVFormer++ [14] | ResNet101-DCN | 0.361 | 0.522 | 0.481 |
| MV-FCOS3D++ [29] | ResNet101-DCN | 0.379 | 0.522 | 0.484 |
| PETR [20] | ResNet101 | 0.358 | 0.502 | 0.462 |
| PETRv2 [21] | ResNet101 | 0.366 | 0.519 | 0.479 |
| StreamPETR [28] | ResNet101 | 0.399 | 0.553 | 0.517 |
| **QE-BEV** | ResNet101 | **0.426** | **0.582** | **0.547** |

the NDS score of QE-BEV rises to 57.8, outperforming the 55.8 by SparseBEV. Moreover, the performance of QE-BEV on the nuScenes test dataset is provided in the supplementary material, which also reveals considerable advantages.

As shown in Table 2, in more complex configurations, such as using a ResNet101 backbone and an input size of 1408 × 512, QE-BEV outshines its competitors with an NDS of 61.1, exceeding SparseBEV's 59.2, making it the current leading approach.

QE-BEV consistently maintains high mAP scores, proving its robust object detection capabilities. In terms of True Positive metrics like mATE, mASE, QE-BEV holds its ground well compared to SparseBEV and other competing methods. Moreover, the model also performs well on fine-grained evaluation metrics such as Object Orientation Error (mAOE) and Attribute Error (mAAE). The application of perspective pre-training not only improves nearly all evaluation metrics but also showcases the model's adaptability and flexibility. Table 3 compares QE-BEV with the SOTA methods on the Waymo validation dataset. It also reveals the advantage of QE-BEV, with the highest scores of mAPL, mAPH and mAP.

The advantages of QE-BEV primarily stem from two inherent aspects: Firstly, the design of QE-BEV allows it to better capture long-range dependencies. In 3D object detection, different parts of

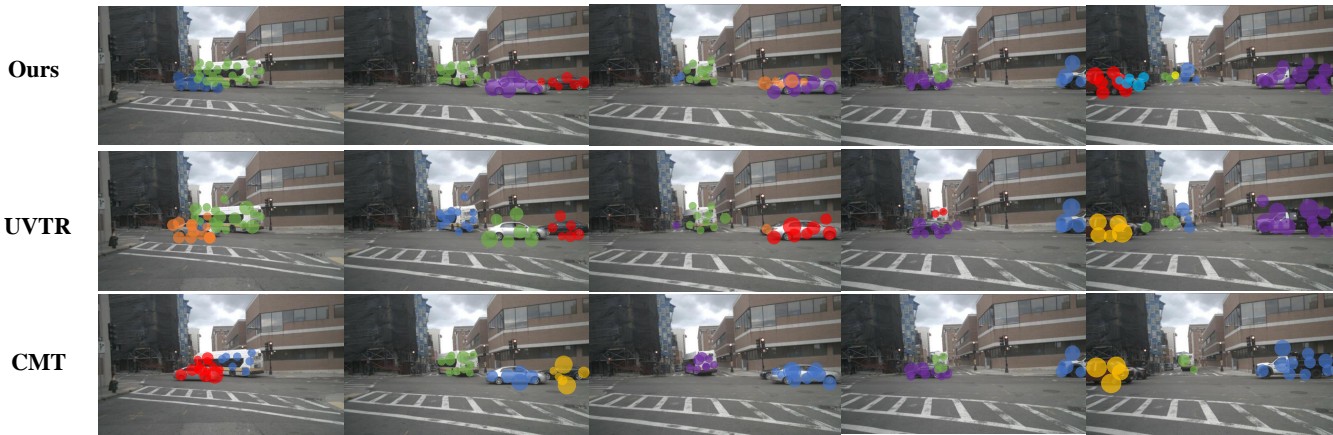

**Figure 4: Comparative visualization of query results for object detection using different dynamic querying methods. Different instances are distinguished by colors. The size of the points indicates depth: larger points are closer to the camera.**

an object might be spatially distant but contextually related. For instance, the front and rear of a car might be far apart in the BEV space, yet they belong to the same object. SparseBEV, being a static query-based method, might struggle in such scenarios since its query points are fixed and cannot dynamically adapt to the changing scene. In contrast, QE-BEV, through its Dynamic Query Evolution Module, can update its query points in real-time, thereby better capturing these long-range dependencies. Secondly, QE-BEV is better equipped to handle the dynamism of real-world scenes. Objects in real-world scenarios might move, rotate, or change their shape. SparseBEV, with its static query points, might falter in such dynamically changing scenes. However, QE-BEV, through its dynamic queries and K-means clustering, can dynamically adjust its query points, thus better adapting to the evolving scene.

## 4.4 Ablation Study

**Dynamic Query Evolution Module (DQEM).** For all ablation studies, we use ResNet-50 as the backbone and adopt the same training and evaluation protocols. The baseline model employs the standard cross-attention mechanism. The Dynamic-K Block integrates Dynamic Queries, K-means Clustering, and Top-K Attention as a unified module. We compare this with the baseline model that uses standard cross-attention.

**Table 4: Ablation study on the Dynamic-K Block.**

| Model Configuration | NDS mAP |
|---|---|
| Baseline (Cross-Attention) | 51.7  40.8 |
| Dynamic-K Block | 56.1  45.4 |

Table 4 shows that the introduction of the Dynamic-K Block results in an 4.2% increase in NDS and a 4.3% increase in mAP compared to the baseline. The Dynamic-K Block's significant performance boost can be attributed to its ability to focus on key features dynamically. Traditional methods with static query points, like the baseline model, might not be able to adapt to the dynamic nature of real-world scenes. In contrast, the Dynamic-K Block, with its integration of Dynamic Queries, K-means Clustering, and Top-K Attention, allows the model to dynamically adjust its focus based on the scene's context. This adaptability ensures that the model can

give precedence to critical features, especially in complex scenes where objects might be occluded or distant from each other.

**Table 5: Ablation study on the Lightweight Temporal Fusion Module (LTFM).**

| Model Configuration | NDS mAP |
|---|---|
| Baseline (No Temporal Fusion) | 52.8  42.3 |
| With LTFM | 56.1  45.4 |
| LSTM-based Fusion | 53.5  43.2 |
| Convolutional LSTM Fusion | 53.7  43.5 |
| Simple Averaging | 52.5  42.0 |

**Lightweight Temporal Fusion Module (LTFM).** To study the effectiveness of our Lightweight Temporal Fusion Module (LTFM), we compare it with the baseline that doesn't employ temporal fusion and other prevalent temporal fusion methods in Table 5. All other configurations remain the same for a fair comparison.

Incorporating the Lightweight Temporal Fusion Module (LTFM) to the baseline model results in a 3.1% increase in NDS and a 2.8% increase in mAP. These improvements indicate that LTFM effectively captures the temporal dependencies without introducing significant computational overhead, thus validating its utility in our QE-BEV framework. The LTFM provides the model with crucial context about these object movements. By fusing information across time, the model gains a more comprehensive understanding of the scene, allowing it to predict object trajectories and interactions more accurately. LTFM consistently outperformed other methods like LSTM-based fusion, Convolutional LSTM fusion, and simple averaging across time. This can be attributed to LTFM's lightweight design and its adeptness at capturing crucial temporal dependencies without significant computational overhead.

We further explored the temporal resolution at which the LTFM operates in Table 6. Different scenarios might benefit from different temporal granularities. When comparing the performance of LTFM at different time intervals, such as every frame, every 2 frames, and every 5 frames, we observed that fusing information at every 2 frames provided the optimal balance between computational efficiency and detection accuracy.

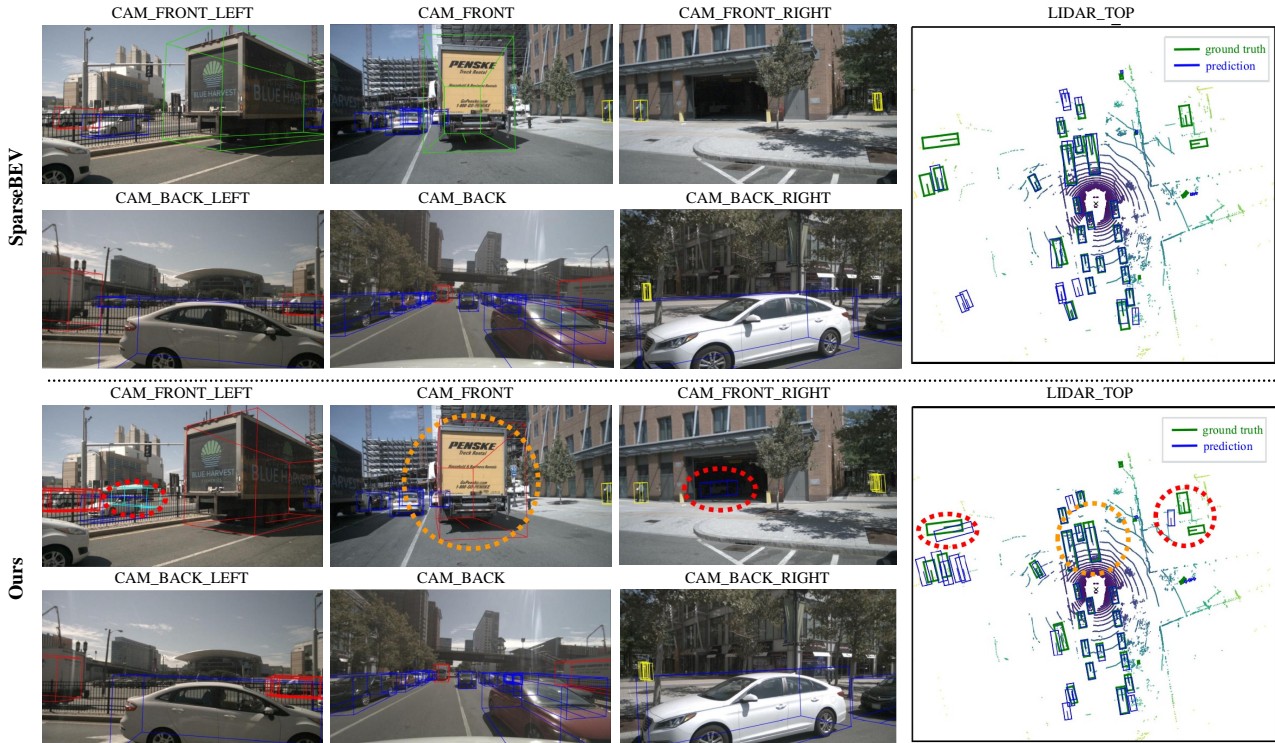

**Figure 5: Visualization 3D object detection results. Detected objects are highlighted with bounding boxes in the camera views and corresponding position markers in the LiDAR top view.**

**Table 6: Performance of LTFM at different temporal resolutions.**

| Temporal Resolution | NDS | mAP |
|---|---|---|
| Every Frame | 55.5 | 44.8 |
| Every 2 Frames | 56.1 | 45.4 |
| Every 5 Frames | 55.2 | 44.5 |

**Selection of $K$ in K-means and Top-K Attention.** As illustrated in Figure 6a, increasing the number of clusters $K$ initially improves both NDS and mAP. The performance plateau observed after $K = 6$ in K-means clustering suggests that there's an optimal number of clusters that capture the scene's essence. Having too many clusters might over-segment the data, leading to redundant or even conflicting information. Similarly, Figure 6b shows that utilizing Top-K Attention with $K = 6$ yields the best performance, highlighting the importance of selective attention. Including Diversity Loss improves both NDS and mAP, as shown in the supplementary material, indicating its effectiveness in balancing the attention mechanism and capturing a variety of features.

### 4.5 Visualization

Comparative visualization of query results for object detection is shown in Figure 4. Our dynamic querying scheme is more adaptive to occlusions and rapidly moving objects, and aligns well across different timestamps. As can be seen from Figure 5, although some targets are difficult to distinguish or locate and are misidentified by SparseBEV, QE-BEV is able to correctly detect the targets. This result confirms the excellent performance of the QE-BEV detector.

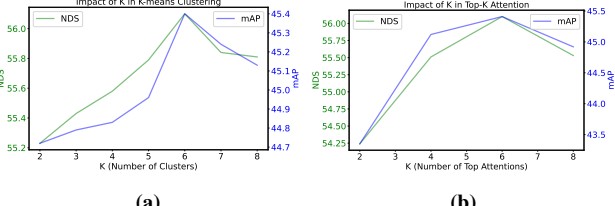

**(a)**      **(b)**

**Figure 6: Performance impact of different parameter settings in K-means and Top-K Attention.**

### 5 Conclusion

In this paper, we presented QE-BEV, a novel approach to 3D object detection that leverages dynamic queries in BEV space. Distinct from conventional static query-based techniques, QE-BEV iteratively adapts queries to capture complex spatial and temporal relationships within the data. This dynamic paradigm offers a more flexible and adaptive mechanism for 3D object detection, effectively constituting a new frontier in the field. Our method integrates various novel components, including K-means clustering for feature selection, Top-K Attention for adaptive feature aggregation, and a Lightweight Temporal Fusion Module for efficient temporal context integration. These components collectively enable our model to outperform state-of-the-art methods on various benchmarks, thus validating the efficacy of the dynamic query-based paradigm.

As future work, we aim to explore the applicability of dynamic queries in other vision tasks and to further optimize the computational efficiency of our model. We also plan to investigate the potential of incorporating more advanced temporal models to capture long-term dependencies in videos or large-scale 3D scenes.

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
