# OpenReview forum: "QE-BEV: Query Evolution for Bird's Eye View Object Detection in Varied Contexts"
_acmmm.org/ACMMM/2024/Conference — MM2024 Poster_

### Official Review · Reviewer_2ugj · 2024-05-23

**Rating:** 5
**Confidence:** 3

**Summary:**

This paper presents a novel method, i.e. QE-BEV, for 3D object detection that leverages dynamic queries in BEV space. QE-BEV distincts previous methods mainly by associating spatial and temporal relationship with the queries. The main components within QE-BEV utilize K-means, Top-K attention and dynamic adaptation. The proposed method achieves competitive resulting on existing 3D object detection tasks.

**Strengths:**

(1)The proposed dynamic query mechanism enables to integrate temporal information from prior implementations.
(2)The proposed Dynamic Query Evolution Module (DQEM) exploits K-means clustering to extract adaptive structure representation and Top-K Attention for refining queries with their associated feature clusters.
(3)The proposed Lightweight Temporal Fusion Module (LTFM) enables to efficiently capture temporal context for each query with attention mechanism.
(4)The overall architecture of the method proposed in the paper is concise and easy to implement, while also achieving the state-of-the-art results on public datasets.

**Limitations:**

(1)It is suggested that the authors to add a comparison between method calculation consumption and model parameter quantity in the experimental section, as the method proposed in the article has the characteristic of lightweight.
(2)It is suggested that the authors add some visualization of intermediate results in the experimental section to better illustrate the role of the module.

**Suitability:**

3

---

### Official Review · Reviewer_Uy4t · 2024-05-24

**Rating:** 4
**Confidence:** 2

**Summary:**

The paper "QE-BEV: Query Evolution for Bird’s Eye View Object Detection in Varied Contexts" introduces a dynamic query evolution framework for 3D object detection in BEV images, crucial for autonomous driving and robotics. The model uses K-means clustering, Top-K attention, and a Lightweight Temporal Fusion Module (LTFM) to improve adaptability and efficiency.

**Strengths:**

1. Innovative dynamic query mechanism for complex scene understanding.

2. Improved computational efficiency through LTFM.

3. Performance gains over existing methods.

**Limitations:**

1. Could you please provide more explanation on the performance degradation of clustering too much i.e. K increase.

2. The paper would be strengthened by a more thorough discussion of the limitations and potential failure cases of the methodology. It is important for readers to understand the circumstances under which the approach may not be effective or could encounter issues, and how these might be addressed or mitigated.

**Suitability:**

2

---

### Official Review · Reviewer_R48b · 2024-06-01

**Rating:** 4
**Confidence:** 3

**Summary:**

The paper proposes QE-BEV, a novel dynamic query mechanism for Bird’s Eye View (BEV) 3D object detection in varied contexts. This method utilizes K-means clustering and Top-K attention mechanisms for refined spatio-temporal data processing, enabling real-time, focused scene analysis. Extensive evaluations on the nuScenes and Waymo datasets demonstrate significant improvements in detection accuracy with QE-BEV.

**Strengths:**

1. The introduction of the dynamic query evolution strategy effectively combines K-means clustering and Top-K attention mechanisms, enhancing the model's adaptability and computational efficiency.

2. Through dynamic querying and the Lightweight Temporal Fusion Module (LTFM), the model efficiently integrates spatio-temporal context, reducing computational overhead.

**Limitations:**

1. Figure 3 is not referenced in the text and is difficult to understand. The authors need to explain it in the corresponding section of the paper.

2. The authors should explain why they chose Top-K attention over other attention mechanisms, such as deformable attention.

3. The authors need to conduct more ablation studies on hyperparameter selection to improve the completeness of the paper. For example, the hyperparameter Lambda for the Diversity Loss (Ldiv) and the choice of the number of queries should be demonstrated.

4. The comparison in Table 1 does not effectively demonstrate the validity of QE-BEV. Firstly, the authors do not mention the reasons behind the design of relevant hyperparameters, such as the number of queries, which significantly impact both efficiency and accuracy. Secondly, the authors only present the FPS for each model but do not provide corresponding performance metrics (NDS, mAP). It is suggested that the authors provide these performance metrics alongside FPS to offer a more comprehensive comparison.

5. The authors need to unify the naming of the model components. For instance, the term "Dynamic-K Block" mentioned in Section 4.1.1 should be clarified as DQEM if they are the same.

**Suitability:**

2

---

### Official Review · Reviewer_3R1q · 2024-06-01

**Rating:** 3
**Confidence:** 3

**Summary:**

This paper introduces QE-BEV, a novel method for improving 3D object detection in bird's-eye view (BEV) images, especially useful in autonomous driving and robotics. QE-BEV utilizes dynamic query evolution and a lightweight temporal fusion module, employing K-means clustering and Top-K attention mechanisms to enhance detection accuracy and efficiency. The method shows significant improvements over existing techniques on the nuScenes and Waymo datasets.

**Strengths:**

1. Dynamic Query Evolution: Introduces a strategy for iteratively updating queries during detection, allowing the model to adapt to complex spatio-temporal relationships, enhancing flexibility and adaptability.

2. Efficient Feature Aggregation: Utilizes K-means clustering and Top-K attention mechanisms to efficiently aggregate features, ensuring the model focuses on the most relevant feature clusters, improving detection accuracy.

**Limitations:**

1. The log file in the supplementary materials contains the author's name, violating the double-blind requirement.
2024-03-24 20:29:36,354 - mmdet - INFO - Start running, host: lizhiqi@HOST-10-142-40-14, work_dir: /home/jiawei/QEBEV/work_dirs/bevformer_Res50

2. Insufficient motivation: The authors do not provide an adequate introduction to static and dynamic query-based methods.

3. Writing: The writing needs improvement. In Figure 1, what do the tokens represent? Is K-means applied to the image features?

4. Experimental section: The experimental results lag behind current methods. The authors lack a comparison of single-frame experimental results with other methods.

**Suitability:**

1

---

### Meta-Review · Area_Chair_1ipH · 2024-06-30

**Recommendation:** Accept (Poster)
**Confidence:** 5

**Metareview:**

This paper proposes a framework utilizing dynamic query evolution strategy in BEV object detection. The problem is that previous implementations of dynamics queries does not leverage temporal information in a computationally efficient manner. Thus, this paper adopt K-means clustering and Top-K attention mechanisms for refined spatio-temporal data processing. Experiments on the nuScenes and Waymo datasets demonstrate its effectiveness.

The preliminary ratings are (br, ba, ba, wa). Key concerns are:
1. Insufficient motivation without an adequate introduction to static and dynamic query-based methods and the reason of the design.
2. Some writing and format issues.
3. Missing some fair comparison to current methods, and the performance is not sound in supporting the claim.
4. Missing analysis in the limitation and the failure cases.

After author rebuttal, the final ratings are (na, na, ba, wa), na for not-available. Reviewer 3R1q and R48b haven't provided their final rating. Reviewer Uy4t and 2ugj maintain their scores after the rebuttal, as the concerns 3&4 are resolved. However, concerns 1&2 still stay. AC stepped in, read the paper, review comments, the rebuttal and the reviewer's final justification. In the view of AC, concern 1 are sufficiently justified in the author's rebuttal by more ablation experiments and comparison to current methods. Given the absence of reviewers' feedback, AC maintain this viewpoint as it is. Concern 2 is viable given enough time to revise the paper.

Based on the comments above, , AC decides to accept the paper since the paper shows the effectiveness of the novel modules under fair setting. The authors should improve the manuscripts according to the comments above.